# Prognostic impact of suspicious extraabdominal lymph nodes on patient survival in advanced ovarian cancer

Kena Park[1,2], Ji Young Kwon[1,2], Jeong Min Song[1,2], Seung Yeon Pyeon[1], Seon Hwa Lee[3], Young Shin Chung[1], Jong-Min Lee[1]*

**1** Department of Obstetrics and Gynecology, Kyung Hee University School of Medicine, Kyung Hee University Hospital at Gangdong, Seoul, Korea, **2** Department of Medicine, Graduate School of Medicine, Kyung Hee University, Seoul, Korea, **3** Research Institute of Clinical Medicine, Medical Big Data Research Center, Kyung Hee University Hospital at Gangdong, Seoul, Korea

* kgo02@hanmail.net

## Abstract

### Objective

To evaluate the clinical impact of suspicious extra-abdominal lymph nodes (EALNs) identified preoperatively on CT and/or PET/CT images in advanced ovarian cancer.

### Methods

A retrospective study was conducted with 122 patients diagnosed with stage III or IV ovarian cancer with preoperative CT and/or PET/CT images from 2006 to 2022. Imaging studies were evaluated for the presence, size and location of suspicious EALNs. Suspicious lymph node enlargement was defined by a cut-off ≥5mm short-axis dimension on CT and/or lesions with maximum standardized uptake values of ≥2.5 on PET/CT. This study only included patients who did not have their EALNs surgically removed.

### Results

A total 109 patients met the inclusion criteria; 36 (33%) had suspicious EALNs and were categorized as "node-positive". The median overall survival (OS) was 45.73 months for the "node-positive" and 46.50 months for the "node-negative" patients (HR 1.17, 95% CI 0.68–2.00, p = 0.579). In multivariate analysis, after adjusting for other variables selected by process of backward elimination using a significance level of p<0.20, suspicious EALNs still showed no clinical significance on OS (aHR 1.20, 95% CI 0.67–2.13, p = 0.537) as well as progression-free survival (aHR 1.43, 95% CI 0.85–2.41, p = 0.174). Old age (aHR 2.23, 95% CI 1.28–3.89, p = 0.005) and platinum resistance (aHR 1.92, 95% CI 1.10–3.36, p = 0.023) affects adversely on OS.

### Conclusion

Suspicious EALNs did not worsen the prognosis of patients with advanced ovarian cancer. However, its impact on survival is not yet clarified. Further investigation is required to assess the clinical significance of suspicious EALNs on preoperative imaging studies.

**Data Availability Statement:** All dataset files are available from the Figshare database (accession number(s) 10.6084/m9.figshare.24720279).

**Funding:** The author(s) received no specific funding for this work.

## Introduction

The incidence of ovarian cancer has been steadily increasing in Korea [1]. Despite easy access to screening tests, such as transvaginal ultrasounds and blood tests, a significant number of ovarian cancer cases are diagnosed at an advanced stage due to minimal and nonspecific symptoms [2]. Chemotherapy following primary cytoreductive surgery, or neoadjuvant chemotherapy followed by interval debulking surgery are the treatments of choice for advanced ovarian cancer; as high as 90% of treated patients show clinical response with combination of paclitaxel and platinum. The extent of residual disease after primary surgery has been known as one of the independent prognostic variables. The achievement of complete tumor resection is, therefore, the most important prognostic factor of progression-free survival (PFS) and overall survival (OS). Thus, the resection of any suspicious extraabdominal lymph nodes (EALNs) has been suggested [3–5]. The introduction of and significant advancements in novel pharmacological approaches such as new cytotoxic agents, anti-vascular endothelial growth factor (anti-VEGF) monoclonal antibody and poly ADP ribose polymerase (PARP) inhibitors, however, not only have improved patient outcomes but also have shifted the perception that these surgically treated cancers can possibly be managed as chronic diseases.

The International Federation of Obstetrics and Gynecology (FIGO) stage reflects the extent of disease and predicts the corresponding prognosis of each patient. According to the 2014 FIGO staging system for ovarian and peritoneal cancers, EALNs metastases are classified as stage IVB [5]. Complex afferent lymphatic drainage from different regions of the upper abdomen and lower mediastinum induces a metastatic spread route, which enlarges the cardiophrenic (CPLNs) and supraclavicular lymph nodes in many abdominopelvic and thoracic malignancies [6,7]. Moreover, lymphatic pathways going through broad and proper ovarian ligaments toward the iliac artery or round ligament of the uterus toward the inguinal region may result in the enlargement of inguinal lymph nodes [8]. Recent reports indicate that enlarged EALNs suspected for malignancy worsen survival in advanced ovarian cancer. As such, timely resection of such EALNs has been proposed [7,9,10].

Previous randomized trials for advanced ovarian cancer have reported that systematic intraabdominal lymphadenectomy was only associated with prolonged PFS but not OS [11]. Additional studies have reported a higher incidence of postoperative complications without improved oncological outcome following systematic pelvic and paraaortic lymphadenectomy [12]. The clinical impact of suspicious EALNs in patients with advanced ovarian cancer is currently limited. Our group has already reported the feasibility of video-assisted thoracoscopic surgery (VATS) in resecting suspicious CPLNs to minimize residual disease and pathologically confirm the FIGO stage [13]. However, the inconvenience, potential risk of morbidity, and uncertainty in therapeutic efficacy of the procedure raise doubt on whether systematic removal of suspicious EALNs should be a routine part of maximal cytoreductive surgery [14–16]. The prognostic significance of suspicious EALNs identified preoperatively on computed tomography (CT) and/or positron emission tomography (PET/CT) images should be evaluated before discussing the necessity of surgical resection of such EALNs.

Therefore, the aim of this study was to determine whether the presence of suspicious EALNs on baseline CT and/or PET/CT images in patients undergoing primary cytoreduction for advanced staged ovarian cancer affects overall and progression-free survivals.

## Materials and methods

### Ethics statement

The Institutional Review Board of the Kyung Hee University Hospital at Gangdong approved our retrospective study and issued a waiver of informed consent (KHNMC 2022-09-050). Data

collection in this study was commenced on October 17, 2022. Maintaining the privacy and security of patient data was a priority during or after data collection.

## Study design

Our study identified 122 patients diagnosed with stage III or IV ovarian cancer during the period from June, 2006 to September, 2022, who met the following inclusion criteria: patients who underwent primary or interval debulking surgery at our institution; patients with baseline abdominopelvic and chest CT or PET/CT imaging to evaluate the presence, size, and location of suspicious EALNs; and patients with available follow-up data. The stage of each patient was determined using the 2014 FIGO classification. Patients who underwent surgical removal of suspicious EALNs (n = 4) and those with distant metastasis to the liver and spleen (n = 1) or follow-up loss (n = 8) were excluded.

Data collection in this study included various clinical, surgical, pathological, and follow-up parameters pertaining recurrence or survival. Important prognostic factors, such as age, body mass index (BMI), initial cancer antigen 125 (CA-125) level, histologic type and grade, intraabdominal largest residual tumor volume after cytoreductive operation, platinum resistance in the first or second line chemotherapy, and BRCA mutation status, were reviewed from medical archives. "No residual disease" indicated that complete macroscopic tumor resection was achieved at abdominal surgery. Each surgical procedure performed during the operation was assigned a point value based on its complexity. The sum of these scores, ranging from 0 to 18, was recorded and categorized as low (0–3), intermediate (4–7) or high ($\geq$8) [3].

Prior to surgery, all patients had baseline contrast abdominopelvic and chest CT or PET/CT scans following standard institutional protocols. All images were reviewed to evaluate the presence of suspicious EALNs by board-certified radiologists to ensure accuracy and consistency in the assessment.

The short-axis dimension and anatomical position of the largest EALN was measured and recorded. According to the guidelines of the European Society of Urogenital Radiology (ESUR) and previous literature, a cut-off value of $\geq$ 5 mm short-axis dimension for CPLNs, $\geq$ 1 cm for mediastinal lymph nodes such as supraclavicular, subcarinal, subphrenic and hilar lymph nodes, $\geq$ 1.5 cm for inguinal lymph nodes, and $\geq$ 5 mm for internal mammary and supradiaphragmatic lymph nodes was applied to categorize lymph nodes as suspicious [17]. Additionally, all PET images were acquired from the skull base to the midthighs to be evaluated both visually and semi-quantitatively. Lesions with maximum standardized uptake values (SUV) of $\geq$ 2.5 were accepted as suspicious lymph nodes. Patients with such abnormal EALNs were categorized as "node-positive". The presence of pleural effusion, a potential indicator of advanced disease and prognosis, was evaluated via chest radiographic or CT findings, with an optional cytological examination of the pleural fluid.

The primary endpoint of this study was overall survival, defined as the time from histologic diagnosis to death from any cause. Tumor assessment was conducted with a follow-up CT every three to six months. Secondary endpoints included progression-free survival, defined as the time from histologic confirmation to the earliest occurrence of progression on CT imaging.

## Statistical analysis

In this study, various statistical methods were employed to analyze the collected data and identify the absence or presence of significant associations between EALNs and patient outcome. Categorical variables were analyzed using chi-square test or the Fisher's exact test, and continuous variables were analyzed using t-test or the Mann–Whitney U test. These analyses allowed

for comparison of different groups within the patient population and the identification of potential differences in various parameters. The Kaplan–Meier method and log-rank test were used to evaluate and compare survival outcomes. Univariate analyses based on a Cox proportional hazards model were conducted to evaluate individual variables in relation to recurrence and survival. To further explore the impact of suspicious EALNs on survival while accounting for other potential compounding factors, multivariate analysis was performed. Variables that showed significance or had a p-value of <0.20 after multiple linear regression using the backward elimination technique were selected for inclusion in the multivariate model. Statistical analyses were performed using the Statistical Package for the Social Sciences (SPSS), version 24.0 for Windows (IBN Corp, Armonk, NY, USA). A p-value of <0.05 was considered statistically significant.

## Results

Initially, a total of 122 patients were evaluated for enrollment; 109 diagnosed and treated for stage III or IV ovarian cancer at Kyung Hee University Hospital at Gangdong between June, 2006 and September, 2022 were identified and included in the study after exclusion criteria. Details of patient ineligibility were documented and are available in a patient selection flow diagram (S1 Fig).

Among the 109 patients, 36 (33%) had suspicious EALNs upon review of baseline imaging. The mean short-axis diameter of enlarged lymph nodes was 16.1 mm, with sizes ranging from 5 to 81 mm. Of those classified as having suspicious EALNs, the most common site of involvement was CPLN, accounting for 66% of the cases (n = 24). Other sites of suspicious EALNs included not only mediastinal (n = 18) such as supraclavicular (n = 14), subphrenic (n = 1) and hilar (n = 1), but also internal mammary (n = 6), inguinal (n = 2) and axillary (n = 3) lymph nodes. All 109 patients underwent baseline abdominopelvic and chest CT scans; 71 (65%) underwent baseline PET/CT scans. Among those with baseline PET/CT scans, 17 (24%) showed signs of suspicious EALNs.

To gain a deeper understanding of the patient population, a comparison was made between "node-positive" (patients with suspicious EALNs) and "node-negative" (patients without suspicious EALNs) individuals. The clinical and tumor characteristics of the two groups appear to be well balanced, with the exception of neoadjuvant chemotherapy and pleural effusion (Table 1). More specifically, 17 (47.2%) "node-positive" patients received neoadjuvant chemotherapy (NACT), while only 19 (26%) in the "node-negative" group received NACT (p = 0.027). The comorbidity of pleural effusion with suspicious EALNs was observed in 18 (50.0%) patients in the "node-positive" group, while only 11 (15.1%) cases of isolated pleural effusion was observed in the "node-negative" group (p<0.001). The presence of suspicious EALNs did not significantly affect overall survival. The median OS for "node-positive" patients was 45.73 months, while "node-negative" patients had a median OS of 46.50 months [hazard ratio (HR) 1.17, confidence interval (CI) 0.68–2.00, p = 0.579] (Fig 1).

While OS did not show a significant difference between the two groups, "node-negative" patients showed better PFS with marginal statistical difference compared to the "node-positive" cohort (HR 1.62, 95% CI 0.98–2.66, p = 0.061) (S2 Fig). Early platinum resistance in the first or second line of chemotherapy was associated with a significantly higher risk of mortality (HR 1.84, 95% CI 1.08–3.12, p = 0.025) and disease progression (HR 2.17, 95% CI 1.34–3.54, p = 0.002). Age also played a role, with older patients having a higher risk of mortality (HR 1.76, 95% CI 1.06–2.94, p = 0.029). Histology types other than serous and endometrioid carcinoma were associated with a higher risk of mortality (HR 1.91, 95% CI 1.09–3.35, p = 0.023). Positive residual disease, indicating incomplete cytoreductive surgery, was also associated with

**Table 1. Baseline characteristics.**

| Characteristics | Node-negative (n = 73) (%) | Node-positive (n = 36) (%) | P value |
|---|---|---|---|
| Age at diagnosis, n (%) | | | 0.949 |
| <54 years | 34 (46.6) | 17 (47.2) | |
| ≥54 years | 39 (53.4) | 19 (52.8) | |
| BMI at diagnosis, n (%) | | | 0.249 |
| <22 kg/m$^2$ | 32 (43.8) | 20 (55.6) | |
| ≥22 kg/m$^2$ | 41 (56.2) | 16 (44.4) | |
| CA-125 level before surgery, n (%) | | | 0.455 |
| <442 U/mL | 38 (52.1) | 16 (44.4) | |
| ≥442 U/mL | 35 (47.9) | 20 (55.6) | |
| Histology, n (%) | | | 0.165 |
| Serous + Endometrioid | 61 (83.6) | 26 (72.2) | |
| Others | 12 (16.4) | 10 (27.8) | |
| Grade, n (%) | | | 0.125 |
| Well/Moderate | 22 (32.8) | 6 (18.2) | |
| Poorly | 45 (67.2) | 27 (81.8) | |
| Surgical complexity group, n (%) | | | 0.373 |
| Low/ Intermediate | 47 (64.4) | 20 (55.6) | |
| High | 26 (35.6) | 16 (44.4) | |
| Residual disease[1], n (%) | | | 0.068 |
| No | 35 (54.7) | 11 (32.4) | |
| Yes | 6 (9.4) | 6 (17.6) | |
| Platinum resistance[2], n (%) | | | 0.667 |
| No | 38 (55.9) | 18 (51.4) | |
| Yes | 30 (44.1) | 17 (48.6) | |
| Neoadjuvant chemotherapy, n (%) | | | 0.027 |
| No | 54 (74.0) | 19 (52.8) | |
| Yes | 19 (26.0) | 17 (47.2) | |
| BRCA 1/2 status, n (%) | | | 0.763 |
| Wild-type | 22 (73.3) | 16 (69.6) | |
| Mutation | 8 (26.7) | 7 (30.4) | |
| Not available | 43 | 13 | |
| Pleural effusion, n (%) | | | <0.001 |
| No | 62 (84.9) | 18 (50.0) | |
| Yes | 11 (15.1) | 18 (50.0) | |

[1]Intraabdominal residual disease

[2]Platinum-resistant in the first or second line of chemotherapy.

BMI, body mass index; CA-125, cancer antigen-125; FIGO, Federation of Gynecology and Obstetrics.

a higher risk of mortality (HR 1.81, 95% CI 1.06–3.07, p = 0.029). In contrast, a BMI higher than 22 kg/m$^2$ was associated with a significantly lower risk of disease progression (HR 0.60, 95% CI 0.37–0.97, p = 0.038). Other variables, including CA-125 values (HR 0.77, 95% CI 0.47–1.27, p = 0.312), tumor grade (HR 1.09, 95% CI 0.62–1.93, p = 0.758), surgical complexity (HR 0.87, 95% CI 0.50–1.52, p = 0.631), neoadjuvant chemotherapy (HR 0.79, 95% CI 0.45–1.40, p = 0.423), and pleural effusion status (HR 0.64, 95% CI 0.36–1.16, p = 0.140) showed no significant association with either mortality or disease progression in the univariate analyses (Table 2).

In multivariate analyses, the process of backward elimination was repeated until only variables with a p-value of <0.20 remained in order to assess the independent prognostic significance of suspicious EALNs while considering other potential confounding factors (Table 3). After adjusting for the selected variables, suspicious EALNs showed no clinical significance on OS (aHR 1.20,

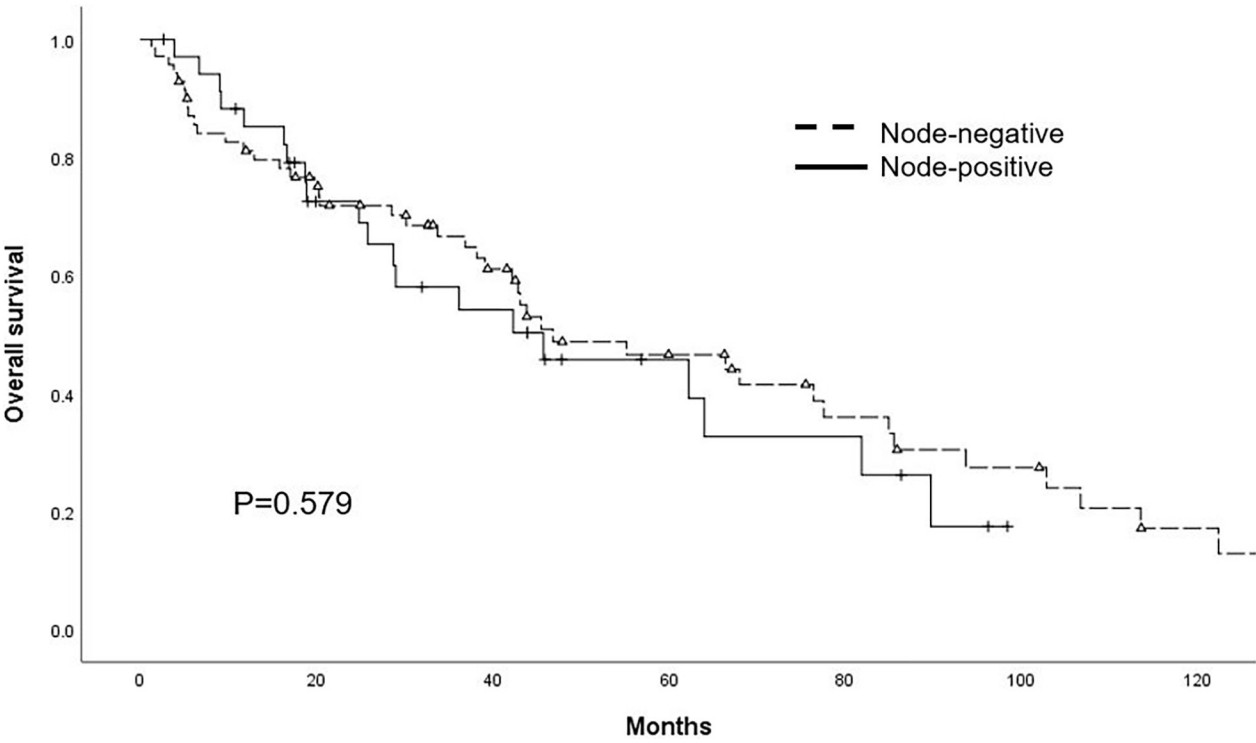

**Fig 1. Kaplan–Meier survival curves of overall survival according to status of extraabdominal lymph nodes.**

95% CI 0.67–2.13, p = 0.537) and PFS (aHR 1.43, 95% CI 0.85–2.41, p = 0.174). Old age (aHR 2.23, 95% CI 1.28–3.89, p = 0.005) and platinum resistance (aHR 1.92, 95% CI 1.10–3.36, p = 0.023) were the two prognostic factors that maintained statistical significance on OS (Fig 2). Platinum resistance adversely affected PFS (aHR 2.51, CI 1.49–4.22, p = 0.001), while neoadjuvant chemotherapy did not improve PFS (aHR 1.85, CI 1.06–3.26, p = 0.032) (Fig 3).

## Discussion

Ovarian, tubal and peritoneal cancers represent a significant challenge in the field of gynecologic oncology due to their advanced stage at diagnosis and complex treatment considerations. The 2014 FIGO staging system has been a crucial framework for categorizing these malignancies, with particular attention paid to the presence of thoracic metastases such as malignant pleural effusion (stage IVA). Lymph node involvement in the thorax or anywhere else outside of the abdominal cavity would lead to an upstaging from FIGO III and IVA to IVB, and this classification may be interpreted as a need for special considerations associated with optimal treatment [6]. However, the prognosis of thoracic involvement, such as enlarged CPLNs, in ovarian cancer patients still remains unclear [18]. This study aimed to investigate the relationship between survival and the presence of suspicious EALNs (including but not limited to CPLNs) in the era of medically-treated chronic illness of ovarian cancer before discussing whether surgical resection of such EALNs is necessary.

### Prognostic implications of suspicious EALNs

Radiological assessment plays a pivotal role in identifying and characterizing suspicious EALNs. Addley and colleagues have made significant contributions to this area by demonstrating the reliability of the ESUR criteria in identifying radiologically abnormal CPLN on CT

**Table 2. Univariate Cox regression analyses of progression-free survival and overall survival of patients with advanced-stage ovarian cancer.**

| Variables | PFS | | OS | |
|---|---|---|---|---|
| | HR (95% CI) | *P*-value | HR (95% CI) | *P*-value |
| Age at diagnosis (yr) | | 0.574 | | 0.029 |
| <54 | 1.00 (Reference) | | 1.00 (Reference) | |
| ≥54 | 0.87 (0.54–1.40) | | 1.76 (1.06–2.94) | |
| BMI at diagnosis (kg/m$^2$) | | 0.038 | | 0.148 |
| <22 | 1.00 (Reference) | | 1.00 (Reference) | |
| ≥22 | 0.60 (0.37–0.97) | | 0.69 (0.42–1.14) | |
| CA-125 before surgery (U/mL) | | 0.939 | | 0.312 |
| <442 | 1.00 (Reference) | | 1.00 (Reference) | |
| ≥442 | 1.02 (0.63–1.64) | | 0.77 (0.47–1.27) | |
| Histology | | 0.156 | | 0.023 |
| Serous + Endometrioid | 1.00 (Reference) | | 1.00 (Reference) | |
| Others | 1.50 (0.86–2.64) | | 1.91 (1.09–3.35) | |
| Grade | | 0.297 | | 0.758 |
| Well/Moderate | 1.00 (Reference) | | 1.00 (Reference) | |
| Poorly | 1.36 (0.77–2.41) | | 1.09 (0.62–1.93) | |
| Surgical complexity group | | 0.504 | | 0.631 |
| Low/Intermediate | 1.00 (Reference) | | 1.00 (Reference) | |
| High | 0.85 (0.52–1.39) | | 0.87 (0.50–1.52) | |
| Residual disease after surgery[1] | | 0.357 | | 0.029 |
| No | 1.00 (Reference) | | 1.00 (Reference) | |
| Yes | 1.25 (0.77–2.03) | | 1.81 (1.06–3.07) | |
| Platinum resistance[2] | | 0.002 | | 0.025 |
| No | 1.00 (Reference) | | 1.00 (Reference) | |
| Yes | 2.17 (1.34–3.54) | | 1.84 (1.08–3.12) | |
| Neoadjuvant chemotherapy | | 0.382 | | 0.423 |
| No | 1.00 (Reference) | | 1.00 (Reference) | |
| Yes | 1.25 (0.76–2.07) | | 0.79 (0.45–1.40) | |
| Pleural effusion | | 0.551 | | 0.140 |
| No | 1.00 (Reference) | | 1.00 (Reference) | |
| Yes | 1.17 (0.70–1.96) | | 0.64 (0.36–1.16) | |

[1]Intraabdominal residual disease

[2]Platinum-resistant in the first or second line of chemotherapy.

BMI, body mass index; CA-125, cancer antigen-125; PFS, progression-free survival; OS, overall survival; HR, hazard ratio; CI, confidence interval.

images [17]. We, therefore, adhered to the 5 mm short-axis diameter cut-off to categorize our patient cohort as "node-positive" or "node-negative". Furthermore, the integration of PET/CT imaging has enhanced accuracy of identifying pathologic CPLNs smaller than 1 cm in size [19]. The combination of anatomical information from CT scans and metabolic data from PET scans provides a more comprehensive evaluation of lymph node involvement. We, therefore, further expanded the definition of "node-positive" to include those with increased uptake on preoperative PET/CT scans.

The central question addressed in this discussion is whether the presence of suspicious EALNs has a significant impact on the prognosis of patients with advanced ovarian cancer, and thus whether surgical resection of such nodes is necessary. Our data suggest that suspicious EALNs do not worsen patient prognosis in advanced ovarian cancer.

An explanation to this insignificance of suspicious EALNs compared to that of intraabdominal lymph nodes on PFS and OS is that bowel obstruction is the most common form of

**Table 3. The selection of potential prognostic variables using backward elimination with a significance level of p<0.20.**

| Variables | PFS | | OS | |
|---|---|---|---|---|
| | aHR (95% CI) | P-value | aHR (95% CI) | P-value |
| | | | | |
| Age at diagnosis (kg/m$^2$) | | | | 0.007 |
| <54 | | | 1.00 (Reference) | |
| ≥54 | | | 2.24 (1.25–4.02) | |
| BMI at diagnosis (kg/m$^2$) | | 0.149 | | 0.167 |
| <22 | 1.00 (Reference) | | 1.00 (Reference) | |
| ≥22 | 0.69 (0.41–1.15) | | 0.67 (0.38–1.18) | |
| Surgical complexity group | | 0.103 | | |
| Low/ Intermediate | 1.00 (Reference) | | | |
| High | 0.63 (0.37–1.10) | | | |
| Platinum resistance[1] | | 0.002 | | 0.064 |
| No | 1.00 (Reference) | | 1.00 (Reference) | |
| Yes | 2.34 (1.37–4.01) | | 1.72 (0.97–3.06) | |
| Neoadjuvant chemotherapy | | 0.003 | | 0.038 |
| No | 1.00 (Reference) | | 1.00 (Reference) | |
| Yes | 2.41 (1.34–4.33) | | 0.48 (0.24–0.96) | |

[1] Platinum-resistant in the first or second line of chemotherapy.

BMI, body mass index; PFS, progression-free survival; OS, overall survival; aHR, adjusted hazard ratio; CI, confidence interval.

tumor-related morbidity in ovarian cancer patients [19]. This underscores the complex nature of ovarian cancer, where symptoms and complications often arise from intraabdominal disease burden. Previous studies have suggested suspicious CPLNs to be a marker for upper abdominal disease, such as thoracic involvement at relapse [20]. However, neoplastic metastases

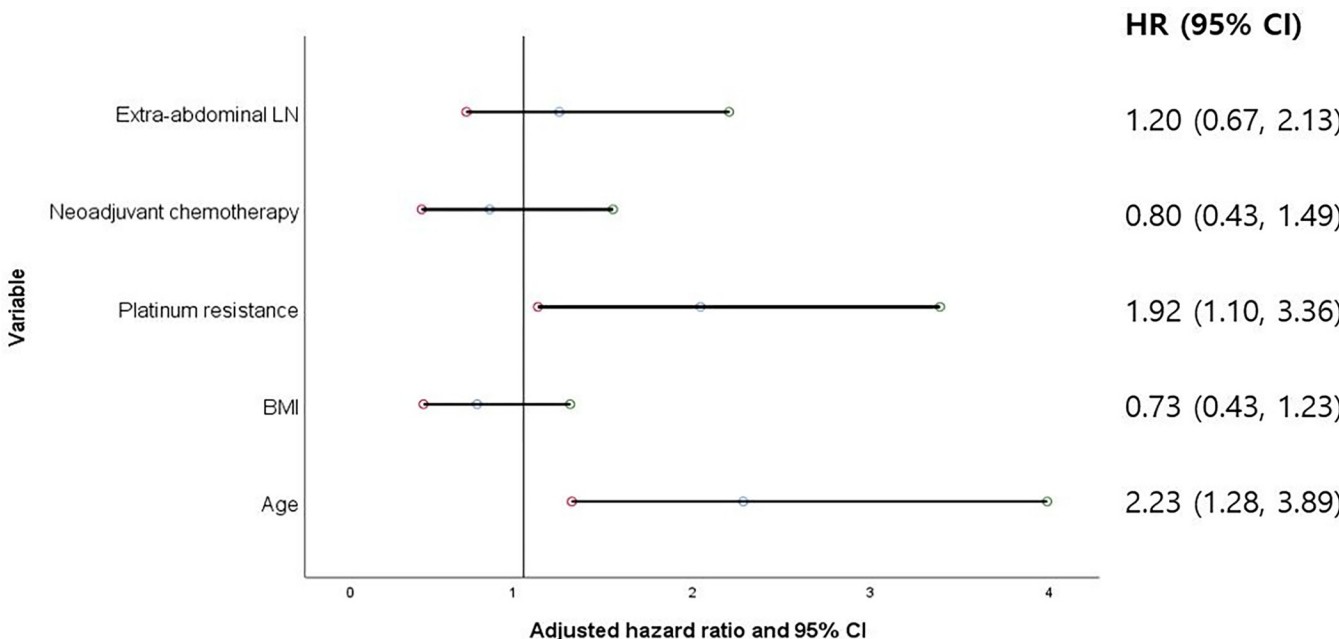

**Fig 2. The significance of extraabdominal lymph nodes on overall survival adjusted by selected covariates, visualized with a forest plot.**

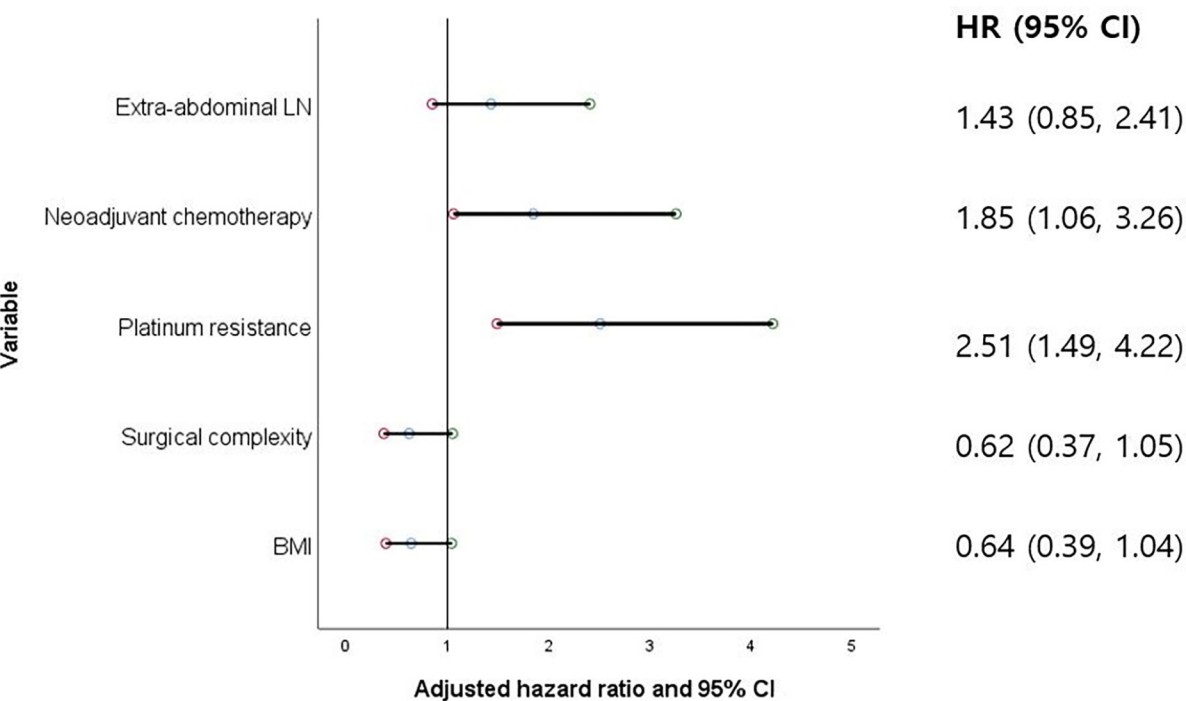

**Fig 3. The significance of extraabdominal lymph nodes on progression-free survival adjusted by selected covariates, visualized with a forest plot.**

associated with mechanical obstruction, and subsequent tumor-induced morbidity occurred mostly in the small and large intestines when multifocal, and in the rectosigmoid region when unifocal. Ureteral obstruction was the second most common cause of death in women with ovarian cancer [19]. Consequently, the clinical significance of thoracic involvement, as represented by suspicious EALNs, may have been masked by the predominant impact of intraabdominal disease on patient outcomes.

Another explanation is the introduction of novel pharmacological approaches that have revolutionized the treatment landscape for ovarian and peritoneal cancers. Newly developed agents such as anti-VEGF monoclonal antibody, PARP inhibitors, and immune checkpoint inhibitors have significantly improved the quality of life and oncologic outcomes for ovarian cancer patients [21]. These advancements have reclassified ovarian cancer, especially in cases with poor prognostic factors such as recurrent disease, stage IV disease and macroscopic residual disease after debulking surgery, into a chronic disease that can be managed conservatively [22]. Clinical trials have demonstrated the benefit of adding bevacizumab to standard chemotherapy regimens in advanced ovarian cancer and platinum-sensitive recurrent settings [23,24]. This has translated into improved PFS for patients. Furthermore, the emergence of PARP inhibitors has brought about a paradigm shift in the treatment of ovarian cancer, particularly in patients with BRCA 1/2 mutations [25–28]. Pembrolizumab monotherapy has been reported to have limited efficacy in the treatment of ovarian cancer [29,30]. Nonetheless, these agents have shown promise and provides new hope for advanced ovarian cancer patients. Newer chemotherapy agents, such as pegylated liposomal doxorubicin, gemcitabine, and topotecan, were also available as alternative options for participants who did not respond to standard treatments or had recurrent disease [31]. Thus, the possibility of novel and improved medical treatments waning the clinical impact of suspicious EALNs cannot be excluded.

## Surgical considerations and therapeutic strategies

Randomized trials determining the efficacy of systematic intraabdominal lymphadenectomy have reported an association with PFS, while additional studies have reported a higher incidence of postoperative complications without improved oncological outcome [12,13,32]. Several groups have investigated the feasibility, safety, and clinical outcomes of CPLNs resection in advanced ovarian cancer via VATS, transdiaphragmatic, or subxiphoid approach to identify and confirm extraabdominal metastases and obtain optimal cytoreduction. Aggressive surgical attempts, such as opening up the diaphragm to enter the thoracic cavity for CPLNs resection, was associated with minimal blood loss and complications. Although these interventions did not worsen morbidity, the surgical gain and therapeutic value still remains uncertain [14,33,34]. Similarly, suspicious EALNs did not show prognostic role on PFS as well as OS in this study. The concept of ovarian cancer transitioning into a medically-treated chronic illness is, once again, a noteworthy development. The advent of novel therapies such as anti-VEGF monoclonal antibodies, PARP inhibitors and immunotherapies may have been waning prognostic significance of suspicious EALNs and raise a question about the therapeutic role of surgical resection of such EALNs.

Instead, platinum resistance and age emerged as critical independent prognostic factors, highlighting the importance of tumor biology and the general condition of the patient.

## Limitations

Although the Korean National Health Insurance system covers a wide range of medical services and procedures, the specific coverage for BRCA testing was initially limited to certain clinical indications, such as strong family history or disease recurrence of breast or ovarian cancer. Over the years, with the accumulation of clinical trial data supporting the positive outcomes for patients with BRCA mutations treated with PARP inhibitors, coverage for not only the BRCA- test but also PARP inhibitor-treatment have been expanded to include more eligible patients. Thus, access to BRCA testing and subsequent treatment with PARP inhibitors has not been uniform among patients, and this explains the notable limitation of the study: insufficient information regarding the BRCA mutation status of the patients. This limitation introduces the possibility of selection bias in the study, as patients with access to these targeted therapies may differ from those without such access, and therefore was not included in further analyses.

Our study was conducted at a single institution, which may limit the generalizability of the findings to broader patient populations. The relatively modest sample size, particularly in the subgroup of patients with suspicious EALNs, is also a limitation that may have influenced the statistical power of the analysis. Larger studies or multicenter collaborations may provide further insights into the prognostic significance of suspicious EALNs.

## Conclusion

Our results indicate that suspicious EALNs did not impact OS as well as PFS. Therefore, the question still remains to whether surgical removal of suspicious EALNs is a viable option in the primary treatment of advanced ovarian cancer as it transitions into a medically-treated chronic illness. The decision to surgically resect suspicious EALNs identified preoperatively on CT and/or PET/CT images should also be made with caution, weighing in the inconvenience and potential risks of undergoing the procedure itself. This, therefore, demands larger cohort studies to further investigate the clinical impact of suspicious EALNs on survival.

## Supporting information

**S1 Fig. Flowchart for the selection of participants.**
(TIF)

**S2 Fig. Kaplan–Meier survival curves of progression-free survival according to status of extraabdominal lymph nodes.**
(TIF)

## Author Contributions

**Conceptualization:** Jong-Min Lee.

**Formal analysis:** Kena Park, Ji Young Kwon, Jeong Min Song, Seung Yeon Pyeon, Seon Hwa Lee.

**Investigation:** Kena Park, Ji Young Kwon.

**Methodology:** Jeong Min Song, Seung Yeon Pyeon, Young Shin Chung, Jong-Min Lee.

**Resources:** Jong-Min Lee.

**Supervision:** Young Shin Chung, Jong-Min Lee.

**Validation:** Seon Hwa Lee, Young Shin Chung.

**Visualization:** Kena Park.

**Writing – original draft:** Kena Park, Jong-Min Lee.

**Writing – review & editing:** Jeong Min Song.

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
