## [Decision Letter · Decision Letter 0]

15 Nov 2023

PONE-D-23-33134Prognostic impact of suspicious extraabdominal lymph nodes on patient survival in advanced ovarian cancerPLOS ONE

Dear Dr. Lee,

Thank you for submitting your manuscript to PLOS ONE. After careful consideration, we feel that it has merit but does not fully meet PLOS ONE’s publication criteria as it currently stands. Therefore, we invite you to submit a revised version of the manuscript that addresses the points raised during the review process.

We look forward to receiving your revised manuscript.

Kind regards,

Federico Romano, M.D., Ph.D.

Academic Editor

PLOS ONE

Journal Requirements:

"This study was partially supported by the research fund from the Sudang Foundation."

Reviewers' comments:

Reviewer's Responses to Questions

**Comments to the Author**

1. Is the manuscript technically sound, and do the data support the conclusions?

Reviewer #1: Yes

Reviewer #2: Partly

2. Has the statistical analysis been performed appropriately and rigorously? 

Reviewer #1: Yes

Reviewer #2: Yes

3. Have the authors made all data underlying the findings in their manuscript fully available?

Reviewer #1: Yes

Reviewer #2: Yes

4. Is the manuscript presented in an intelligible fashion and written in standard English?

Reviewer #1: Yes

Reviewer #2: Yes

5. Review Comments to the Author

Reviewer #1: I read with great interest the Manuscript titled " Prognostic impact of suspicious extraabdominal lymph nodes on patient survival in advanced ovarian cancer”, topic interesting enough to attract readers' attention.

Authors should clarify some point and improve the discussion citing relevant and novel key articles about the topic:

- I suggest a round of language revision, in order to correct few typos and improve readability.

- Considering topic and results if this study, I suggest that authors to add a reference to current evidence about the role of lymphadenectomy in advanced ovarian cancer, considering the advantages and limitations of this procedure. I would be glad if the authors discuss this important point, referring to PMID: 32036457

Considered all this points, I think it could be of interest for the readers and, in my opinion, it deserves the priority to be published after minor revisions

Reviewer #2: It is of great significance to assess the prognostic role of extra-abdominal lymph nodes metastasis in advanced ovarian cancer and weight the advantages and disadvantages of EALNs resection.

Major points:

1. The authors categorized lymph nodes as suspicious, a cut-off value of ≥ 5 mm short-axis dimension, according to the guidelines of the European Society of Urogenital Radiology (ESUR) and previous literature. According to Ref17 Forstner R, Sala E, Kinkel K, Spencer JA. ESUR guidelines: ovarian cancer staging and follow-up. Eur Radiol. 2010;20(12):2773-80, the suspicious lymph nodes were with a short-axis diameter of >1 cm, or suspicious clusters of smaller lymph nodes, or cardiophrenic lymph nodes with a short-axis diameter of >5 mm were considered suspicious. A short-axis diameter of >5mm was only involved in the cardiophrenic lymph nodes in this reference. To the best of my knowledge, the 2022 EUSR (Eur Radiol. 2022 May;32(5):3220-3235) defined lymph nodes metastasis not only considering the short-axis dimension, but also the anatomical position. For example, a short-axis diameter of >1 cm for mediastinal lymph nodes, 1.5 cm for inguinal lymph nodes, and >5 mm for internal mammary and supradiaphragmatic lymph nodes. Since this study included patients with not only supradiaphragmatic lymph nodes, different anatomical information should be considered in the diagnostic criteria.

2. “Lesions with maximum standardized uptake values (SUV) of ≥ 2.5 were accepted as suspicious lymph nodes.” Though the authors combined anatomical information from CT scans and metabolic data from PET to scans provide a more comprehensive evaluation of lymph node involvement, the cut-off of SUV ≥2.5 was not solid enough to differentiate “node-positive” and “node-negative”. By the way, did all the 109 patients receive PET/CT and CT scans? Otherwise, how much percentage of patients receive PET/CT scan and diagnosed as “node-positive”? This kind of information should be listed.

3. “This study only included patients who did not have their EALNs surgically removed.” Since all the suspicious lymph nodes were diagnosed according to imaging scan without pathologically confirmed, the accuracy of diagnosis could be challenged, so as the final conclusion that suspicious EALNs did not worsen the prognosis of patients with advanced ovarian cancer.

4. As the authors mentioned, the sample size might influence the statistical power of the analysis.

Minor points:

5. The authors stated that among the 36 patients who had suspicious EALNs, there were CPLN (n=24), supraclavicular (n=11), internal mammary (n=4), and inguinal lymph nodes (n=1). I was curious that there were no patients with suspicious mediastinal, axillary, or retrocrural lymph nodes?

6. At line 96 and 205, the reference article should be Ref 17?

7. “More specifically, 17 (47.2%) “node-positive patients received neoadjuvant chemotherapy (NACT), while only 19 (26%) in the “node-negative” group received NACT (p=0.027).” Did the imbalance of ratio of NACT between “node-positive” and “node-negative” groups affect the final results?

8. Why was “resistance within third line chemotherapy”, not platinum-resistance in the first or second line, utilized as a factor to do the analysis?

6. PLOS authors have the option to publish the peer review history of their article (what does this mean?). If published, this will include your full peer review and any attached files.

Reviewer #1: No

Reviewer #2: No

---

## [Author Response · Author response to Decision Letter 0]

5 Dec 2023

Reviewer #1: I read with great interest the Manuscript titled " Prognostic impact of suspicious extraabdominal lymph nodes on patient survival in advanced ovarian cancer”, topic interesting enough to attract readers' attention.

Authors should clarify some point and improve the discussion citing relevant and novel key articles about the topic:

- I suggest a round of language revision, in order to correct few typos and improve readability.

- Considering topic and results if this study, I suggest that authors to add a reference to current evidence about the role of lymphadenectomy in advanced ovarian cancer, considering the advantages and limitations of this procedure. I would be glad if the authors discuss this important point, referring to PMID: 32036457.

Considered all this points, I think it could be of interest for the readers and, in my opinion, it deserves the priority to be published after minor revisions

Address: Thank you so much for your comments. 

- The attached manuscript has been proofread by Editage, a professional English language editing and publication support service. However, we adhered to your suggestion of additional language revision to correct typos and improve readability.

- We wholly agreed with you. We have added the suggested reference [32] to current evidence about the role of lymphadenectomy in advanced ovarian cancer in line 275-277. “Randomized trials determining the efficacy of systematic intraabdominal lymphadenectomy have reported an association with PFS, while additional studies have reported a higher incidence of postoperative complications without improved oncological outcome [12, 13, 32].” 

Reviewer #2: It is of great significance to assess the prognostic role of extra-abdominal lymph nodes metastasis in advanced ovarian cancer and weight the advantages and disadvantages of EALNs resection.

Major points:

1. The authors categorized lymph nodes as suspicious, a cut-off value of ≥ 5 mm short-axis dimension, according to the guidelines of the European Society of Urogenital Radiology (ESUR) and previous literature. According to Ref17 Forstner R, Sala E, Kinkel K, Spencer JA. ESUR guidelines: ovarian cancer staging and follow-up. Eur Radiol. 2010;20(12):2773-80, the suspicious lymph nodes were with a short-axis diameter of >1 cm, or suspicious clusters of smaller lymph nodes, or cardiophrenic lymph nodes with a short-axis diameter of >5 mm were considered suspicious. A short-axis diameter of >5mm was only involved in the cardiophrenic lymph nodes in this reference. To the best of my knowledge, the 2022 EUSR (Eur Radiol. 2022 May;32(5):3220-3235) defined lymph nodes metastasis not only considering the short-axis dimension, but also the anatomical position. For example, a short-axis diameter of >1 cm for mediastinal lymph nodes, 1.5 cm for inguinal lymph nodes, and >5 mm for internal mammary and supradiaphragmatic lymph nodes. Since this study included patients with not only supradiaphragmatic lymph nodes, different anatomical information should be considered in the diagnostic criteria.

Address: We wholly agreed with you. For further clarification, we have added suggested details pertaining the diagnostic criteria according to the 2022 guidelines of ESUR of suspicious EALNs in line 128-133. “The short-axis dimension and anatomical position of the largest EALN was measured and recorded. According to the guidelines of the European Society of Urogenital Radiology (ESUR) and previous literature, a cut-off value of ≥ 5 mm short-axis dimension for CPLNs, ≥ 1 cm for mediastinal lymph nodes such as supraclavicular, subcarinal, subphrenic and hilar lymph nodes, ≥ 1.5 cm for inguinal lymph nodes, and ≥ 5 mm for internal mammary and supradiaphragmatic lymph nodes was applied to categorize lymph nodes as suspicious [17].” 

We also thoroughly reviewed our data to confirm that our “node-positive” patients indeed adhered to the mentioned criteria. For example, suspicious mediastinal lymph nodes were detected in 15 of the patients. All were categorized as “node-positive” due to 1) isolated mediastinal lymph node enlargement of short-axis dimension of ≥ 1cm, 2) synchronous cardiophrenic lymph node enlargement of short-axis dimension of ≥ 5mm, or 3) positive PET/CT scans.

2. “Lesions with maximum standardized uptake values (SUV) of ≥ 2.5 were accepted as suspicious lymph nodes.” Though the authors combined anatomical information from CT scans and metabolic data from PET to scans provide a more comprehensive evaluation of lymph node involvement, the cut-off of SUV ≥2.5 was not solid enough to differentiate “node-positive” and “node-negative”. By the way, did all the 109 patients receive PET/CT and CT scans? Otherwise, how much percentage of patients receive PET/CT scan and diagnosed as “node-positive”? This kind of information should be listed.

Address: We wholly agreed with you. We reviewed our data to provide suggested information pertaining baseline CT and/or PET/CT scans in line 174-175. “All 109 patients underwent baseline abdominopelvic and chest CT scans; 71 (65%) underwent baseline PET/CT scans. Among those with baseline PET/CT scans, 17 (24%) showed signs of suspicious EALNs.” We highly appreciate your input on the cut-off value of SUV ≥2.5 being not solid enough to differentiate “node-positive” and “node-negative”; we abided to the criteria followed by board-certified radiologists at our institution to increase uniformity and minimize variability.

3. “This study only included patients who did not have their EALNs surgically removed.” Since all the suspicious lymph nodes were diagnosed according to imaging scan without pathologically confirmed, the accuracy of diagnosis could be challenged, so as the final conclusion that suspicious EALNs was not significantly associated with the poor prognosis of patients with advanced ovarian cancer.

Address: We highly appreciate and wholly agree with your comment pertaining the accuracy of diagnosis being challenged with the lack of pathologic confirmation. However, we deemed the uncertainty of not knowing whether or not suspicious EALNs on baseline CT and/or PET/CT images are indeed malignant a realistic and essential factor that needs to be implemented in this trial. The objective of this study was to evaluate the prognosis of preoperatively detected suspicious EALNs, and without knowing its malignancy or lack thereof, discuss whether resection is necessary.

4. As the authors mentioned, the sample size might influence the statistical power of the analysis.

Address: We highly appreciate and wholly agree with your comment pertaining our sample size and its influence on statistical power of the analysis. However, at our institution alone, we are currently unable to accumulate enough data. We acknowledge the need for larger prospective cohorts to assess the clinical impact of suspicious EALNs identified preoperatively on CT and/or PET/CT images in advanced ovarian cancer.

Minor points:

5. The authors stated that among the 36 patients who had suspicious EALNs, there were CPLN (n=24), supraclavicular (n=11), internal mammary (n=4), and inguinal lymph nodes (n=1). I was curious that there were no patients with suspicious mediastinal, axillary, or retrocrural lymph nodes?

Address: We wholly agreed with you. For further clarification, we have added suggested details pertaining the anatomical position of detected suspicious EALNs in line 172-174. “Other sites of suspicious EALNs included not only mediastinal (n=18) such as supraclavicular (n=14), subphrenic (n=1) and hilar (n=1), but also internal mammary (n=6), inguinal (n=2) and axillary (n=3) lymph nodes.”

6. At line 96 and 205, the reference article should be Ref 17?

Address: We apologize for the mistake of citing the wrong reference. We have revised the reference and have cited the appropriate reference of [17] at line 133 and 237. 

7. “More specifically, 17 (47.2%) “node-positive patients received neoadjuvant chemotherapy (NACT), while only 19 (26%) in the “node-negative” group received NACT (p=0.027).” Did the imbalance of ratio of NACT between “node-positive” and “node-negative” groups affect the final results?

Address: We highly appreciate your comment pertaining the imbalance of neoadjuvant chemotherapy (NACT) ratio between the “node-positive” and “node-negative” groups possibly affecting the final results. This study aimed to determine whether the presence of suspicious EALNs affect oncologic outcomes. Variables that showed significance or had a p-value of <0.20 after multiple linear regression using the backward elimination technique were selected for inclusion in the multivariate model. To further explore the impact of suspicious EALNs on survival while accounting for other potential compounding factors including NACT, multivariate analysis was performed. Thus, we think that the results of this study reflect the effects of NACT.

8. Why was “resistance within third line chemotherapy”, not platinum-resistance in the first or second line, utilized as a factor to do the analysis?

Address: We wholly agreed with you. The term we inaccurately phrased “resistance within third line chemotherapy” due to our language barrier and limitation in vocabulary, was actually depicting “platinum-resistance in the first or second line”. We have revised our manuscript to read the suggested “platinum-resistance in the first or second line” phrase.

---

## [Decision Letter · Decision Letter 1]

7 Feb 2024

Prognostic impact of suspicious extraabdominal lymph nodes on patient survival in advanced ovarian cancer

PONE-D-23-33134R1

Dear Dr. Lee,

We’re pleased to inform you that your manuscript has been judged scientifically suitable for publication and will be formally accepted for publication once it meets all outstanding technical requirements.

Kind regards,

Andrea Giannini

Academic Editor

PLOS ONE

Additional Editor Comments (optional):

The manuscript has been modified with the comments of the reviewers. It is now ready to be published.

Reviewers' comments:

Reviewer's Responses to Questions

**Comments to the Author**

1. If the authors have adequately addressed your comments raised in a previous round of review and you feel that this manuscript is now acceptable for publication, you may indicate that here to bypass the “Comments to the Author” section, enter your conflict of interest statement in the “Confidential to Editor” section, and submit your "Accept" recommendation.

Reviewer #1: All comments have been addressed

Reviewer #3: All comments have been addressed

2. Is the manuscript technically sound, and do the data support the conclusions?

Reviewer #1: Yes

Reviewer #3: Yes

3. Has the statistical analysis been performed appropriately and rigorously? 

Reviewer #1: No

Reviewer #3: Yes

4. Have the authors made all data underlying the findings in their manuscript fully available?

Reviewer #1: Yes

Reviewer #3: Yes

5. Is the manuscript presented in an intelligible fashion and written in standard English?

Reviewer #1: No

Reviewer #3: Yes

6. Review Comments to the Author

Reviewer #1: The quality of the manuscript has improved thanks to the changes made. I think it could be of interest to the readers and, in my opinion, it deserves the priority to be published.

Reviewer #3: (No Response)

7. PLOS authors have the option to publish the peer review history of their article (what does this mean?). If published, this will include your full peer review and any attached files.

Reviewer #1: No

Reviewer #3: No

---

## [Editor Report · Acceptance letter]

16 May 2024

PONE-D-23-33134R1 

PLOS ONE

Dear Dr. Lee, 

I'm pleased to inform you that your manuscript has been deemed suitable for publication in PLOS ONE. Congratulations! Your manuscript is now being handed over to our production team.

Kind regards, 

on behalf of

Dr. Andrea Giannini 

%CORR_ED_EDITOR_ROLE%

PLOS ONE